# Control of TRPM3 Ion Channels by Protein Kinase CK2-Mediated Phosphorylation in Pancreatic β-Cells of the Line INS-1

**DOI:** 10.3390/ijms222313133

**Published:** 2021-12-04

**Authors:** Alexander Becker, Claudia Götz, Mathias Montenarh, Stephan E. Philipp

**Affiliations:** 1Experimental and Clinical Pharmacology and Toxicology, Center for Molecular Signaling (PZMS), Saarland University, Building 46, D-66424 Homburg, Germany; AlexBecker5683@hotmail.de; 2Medical Biochemistry and Molecular Biology, Saarland University, Building 44, D-66424 Homburg, Germany; claudia.goetz@uks.eu (C.G.); mathias.montenarh@uks.eu (M.M.)

**Keywords:** transient receptor potential M 3 channels (TRPM3), protein kinase CK2, calcium, glucose-stimulated insulin release (GSIS), INS-1

## Abstract

In pancreatic β-cells of the line INS-1, glucose uptake and metabolism induce the openings of Ca^2+^-permeable TRPM3 channels that contribute to the elevation of the intracellular Ca^2+^ concentration and the fusion of insulin granules with the plasma membrane. Conversely, glucose-induced Ca^2+^ signals and insulin release are reduced by the activity of the serine/threonine kinase CK2. Therefore, we hypothesized that TRPM3 channels might be regulated by CK2 phosphorylation. We used recombinant TRPM3α2 proteins, native TRPM3 proteins from INS-1 β-cells, and TRPM3-derived oligopeptides to analyze and localize CK2-dependent phosphorylation of TRPM3 channels. The functional consequences of CK2 phosphorylation upon TRPM3-mediated Ca^2+^ entry were investigated in Fura-2 Ca^2+^-imaging experiments. Recombinant TRPM3α2 channels expressed in HEK293 cells displayed enhanced Ca^2+^ entry in the presence of the CK2 inhibitor CX-4945 and their activity was strongly reduced after CK2 overexpression. TRPM3α2 channels were phosphorylated by CK2 in vitro at serine residue 1172. Accordingly, a TRPM3α2 S_1172_A mutant displayed enhanced Ca^2+^ entry. The TRPM3-mediated Ca^2+^ entry in INS-1 β-cells was also strongly increased in the presence of CX-4945 and reduced after overexpression of CK2. Our study shows that CK2-mediated phosphorylation controls TRPM3 channel activity in INS-1 β-cells.

## 1. Introduction

A tight control of the insulin concentration in the blood is essential to maintain blood sugar levels within a narrow range and to prevent diabetes mellitus in humans. β-cells of the pancreas are unique with regard to their production and release of insulin. Their absence is not compensable by other cells and a reason for type 1 diabetes. The main stimulus for the fusion of insulin-containing granules with the β-cell membrane and the release of insulin is the elevation of the Ca^2+^ concentration within the cytosol of the cell ([Ca^2+^]_cyt_) [1,2]. This increase is determined by a number of pathways including Ca^2+^ entry into the cell across the plasma membrane [2]. Among these pathways, the opening of voltage-gated Ca^2+^ channels (Cav) is regarded as the main mechanism for the rise of [Ca^2+^]_cyt_ [1,2]. These channels are highly permeable for Ca^2+^ but almost impermeable for monovalent cations allowing a selective inflow of Ca^2+^ even under physiological ion conditions [3]. The opening of Cav channels is triggered by voltage changes, i.e., the depolarization of the membrane potential. Following the uptake and metabolism of glucose and the production of ATP, this membrane depolarization can be evoked by the closure of ATP-dependent potassium channels. However, any ion motion across the plasma membrane affecting the membrane potential could have an effect on [Ca^2+^]_cyt_ and, thus, on insulin secretion [1].

We recently showed that mice lacking transient receptor potential melastatin 3 (TRPM3) proteins display impaired glucose tolerance after intraperitoneal and oral administration of glucose, suggesting an essential role of TRPM3 in insulin release [4]. TRPM3 proteins build Ca^2+^ permeable channels [5,6,7] that can be activated by the neurosteroid pregnenolone sulfate (PS, [8]) as well as the synthetic ligand CIM0216 [9]. These agonists open two alternative ion permeation pathways that are determined by the length of the pore loop of the participating TRPM3 isoforms [9,10]. Whereas PS is an effective agonist of short pore isoforms, long pore loop variants are insensitive to PS but sensitive to CIM0216 [10]. The activity of TRPM3 channels is enhanced by membrane phosphoinositides [11,12] and inhibited by Ca^2+^/calmodulin [13,14]. They belong to the TRP family of cation channels encompassing 28 members in mammals and subdivided into 6 subfamilies (TRPC1–7, TRPV1–6, TRPM1–8, TRPA1, TRPP2,3,5, and TRPML1–3 [15]). In neurons, where their activity is tightly controlled by the Gβγ-subunits of G-protein coupled receptors [16,17,18,19], they serve as temperature-dependent pain sensors [20]. TRPM3 channels are also expressed in primary pancreatic β-cells [8]. There, they build an influx pathway for Zn^2+^ ions that are co-released with insulin [21] and they enhance insulin release in response to PS [8,9]. In β-cells of the line INS-1, TRPM3 channels get also directly activated by elevated levels of glucose leading to increased [Ca^2+^]_cyt_ and insulin release [4]. In addition, TRPM3 channels trigger the activity of Cav channels, so that both channels synergistically increase [Ca^2+^]_cyt_ and insulin release [4]. Recently, we found [Ca^2+^]_cyt_ and insulin release strongly increased after inhibition of the serine/threonine kinase CK2 with CX-4945 or quinalizarin, and we identified Cav2.1 channels as a substrate of CK2 phosphorylation [22], suggesting a modulating role of CK2-mediated Cav channel phosphorylation in insulin release.

Similar to Cav channels, TRPM3 channels contain a number of putative target sites for CK2 phosphorylation, and we hypothesized that CK2 phosphorylation may control the activity of TRPM3 channels, too. Here, we show for the first time that the activity of TRPM3 channels depends on phosphorylation. Recombinant TRPM3α2 channels as well TRPM3 channels endogenous to INS-1 β-cells display enhanced activity after CK2 inhibition and reduced activity after CK2 overexpression. TRPM3α2 channels are phosphorylated in vitro by CK2 at serine residue 1172. This phosphorylation significantly reduces TRPM3-mediated Ca^2+^ entry and might, therefore, contribute to the TRPM3-mediated control of insulin release.

## 2. Results

### 2.1. CK2 Phosphorylates TRPM3 

First, we tested whether CK2 phosphorylates TRPM3 proteins in vitro. We chose the TRPM3 isoform TRPM3α2, which appeared to be most abundant in INS1 cells [4]. HEK293 cells, which do not express TRPM3 endogenously [5], were transfected with the TRPM3α2 cDNA. Three days later, proteins were precipitated using monoclonal anti-TRPM3 antibodies and incubated with [γ-^32^P]ATP in the presence or absence of added recombinant CK2 (Figure 1A). Likewise, we detected strong phosphorylation of ~200 kD proteins in TRPM3-transfected HEK293 cells that were not detected in HEK control cells or in the absence of CK2. The data show that TRPM3 proteins get phosphorylated by CK2 in vitro.

However, after longer exposure of the autoradiograph, we got a faint signal from proteins of the same molecular mass even without CK2 addition (not shown), indicating that kinases endogenous to HEK cells associate with and phosphorylate TRPM3. To analyze whether the associated kinase is indeed protein kinase CK2, we repeated the experiments under conditions that were favorable for CK2. CK2 is one of the rare kinases that can use GTP as phosphate donor with nearly the same efficiency as ATP [23]. We immunoprecipitated proteins from HEK293 or HEK293 cells of the line HEKα2 that stably overexpress TRPM3α2 proteins [24] either with mono- or polyclonal TRPM3 antibodies and incubated the immunoprecipitates with [γ-^32^P]GTP. With both anti-TRPM3 antibodies, we observed in immunoprecipitates of HEKα2 but not HEK293 control cells a phosphorylated protein with a molecular mass of ~ 200 kD (Figure 1B,C). We conclude that kinases endogenous to HEKα2 cells associate with TRPM3 channels and co-immunopurify together with TRPM3. These kinases can use GTP for phosphorylation. 

In order to strengthen the evidence that the associated kinase is protein kinase CK2, immunoprecipitated proteins were subjected after incubation with GTP to an immunoblot analysis using phosphospecific antibodies, which recognize a CK2 substrate motif in its phosphorylated form. We observed a distinct signal at the appropriate molecular mass for TRPM3 only in immunoprecipitates of HEKα2 cells (Figure 1D). Thus, the co-immunoprecipitated kinase can use GTP and phosphorylates TRPM3 at consensus sequences, which are typical motifs for the acidophilic kinase CK2. We suggest that the associated kinase is protein kinase CK2.

### 2.2. CK2 Phosphorylation Inhibits TRPM3-Mediated Ca^2+^ Entry

Does phosphorylation change TRPM3-driven processes? To answer this question, we analyzed HEKα2 cells in Fura-2 Ca^2+^ imaging experiments. To block the endogenous CK2 present in these cells [25], we used the CK2 antagonist CX-4945. Addition of 10 µM CX-4945 directly to HEKα2 cells did not change the intracellular Ca^2+^ concentration (data not shown), demonstrating that CX-4945 has no immediate agonistic effect on TRPM3 channels or other calcium entry pathways in HEK293 cells. Then, we pre-incubated the cells for 30 min in 10 µM CX-4945, washed them twice, and analyzed them before and after stimulation with the TRPM3 agonists pregnenolone sulfate (PS) and CIM0216. We found the TRPM3-mediated Ca^2+^ entry strongly enhanced when the cells were pretreated with the CK2 inhibitor, indicating that CK2-dependent phosphorylation inhibits TRPM3α2 channels (Figure 2A). To confirm that finding, we analyzed HEKα2 cells overexpressing CK2α. Cells were transfected with cDNA encoding fusion proteins consisting of the enhanced green fluorescent protein (EGFP) and CK2α. Green fluorescent cells were compared to non-fluorescent neighboring cells of the same dish and to non-transfected cells (Figure 2B). Introduction of recombinant CK2α reduced the PS/CIM0216-induced increase of [Ca^2+^]_cyt_ significantly, demonstrating again that CK2 phosphorylation modulates TRPM3 channels.

### 2.3. Identification of CK2 Phosphorylation Sites in TRPM3 Proteins

TRPM3 proteins contain 32 amino acid motifs of the sequence S/T XX D/E, in which either a serine or a threonine residue could be a target of CK2 phosphorylation (Figure 3A). We systematically analyzed these putative phosphorylation sites within the complete murine TRPM3 amino acid sequence (including all 28 protein-coding exons) by in vitro phosphorylation of a TRPM3 peptide library (Figure 3B). For that purpose, pentadecapeptides containing the putative serine/threonine targets in their center were synthesized on cellulose membranes. As controls, corresponding peptides, carrying non-phosphorylatable alanine residues instead, were synthesized next to them. Control peptides of the sequence RRRDDDSDDD (positive control) and RRRDDDADDD (negative control) were also included. Filters were incubated with radiolabeled [γ-^32^P]ATP in the presence or absence of recombinant CK2. The positive control peptide displayed strong radiolabeling that was absent when serine residues were replaced with alanine (Figure 3B). 

An equally strong phosphorylation exhibited a TRPM3 peptide with the sequence CRWRKHES_1280_DQDERDY containing the residue serine 1280 (S_1280_). This signal was absent in the control peptide with S_1280_ replaced by alanine indicating specific phosphorylation of S_1280_. The peptides containing the residues S_124_, T_1395_, S_1765_, and S_1789_ displayed phosphorylation signals, too. However, the corresponding control peptides displayed similar signals, indicating non-specific phosphorylation of non-canonical phosphorylation sites within the sequences or unspecific binding of [γ-^32^P]ATP. We found weak phosphorylation of the pentadecapeptide EKDDRFNS_1322_S_1323_NDERIR, which disappeared, however, as soon as one of the two serine residues was mutated. In summary, at least in experiments using in vitro phosphorylation of pentadecapeptides, S_1280_ appeared as the only single TRPM3 residue that was specifically phosphorylated by CK2.

### 2.4. Phosphorylation of S_1280_ Reduces TRPM3 Channel Activity

We next asked whether the phosphorylation of S_1280_ may change TRPM3 channel activity. To answer this question, we mutated the corresponding serine residue S_1172_ of the most common isoform TRPM3α2 to alanine and compared the Ca^2+^ entry through the mutant channel TRPM3α2 S_1172_A to that through the wild-type channel. We did not detect any difference of expression between the wild-type and the mutated TRPM3α2 protein (Figure 4A). However, we observed an increased [Ca^2+^]_cyt_ in HEK293 cells transfected with the mutant already before stimulation of the channels, indicating an increased basal activity of TRPM3α2 channels in the absence of phosphorylation of amino acid residue 1172 (Figure 4B). After full stimulation of the channels with the TRPM3 agonists, PS and CIM2016, the difference in Ca^2+^ entry between mutant and wild-type-expressing cells was strongly enhanced showing that phosphorylation of S_1172_ in the TRPM3α2 variant controls its maximal activity. We conclude that phosphorylation of S_1280_ is a regulatory mechanism to limit basal and maximum TRPM3-mediated Ca^2+^ entry.

### 2.5. CK2 Controls TRPM3 Channels in INS-1 β-Cells

The data shown so far describe the interplay of CK2 and recombinant TRPM3α2 channels. To analyze the influence of CK2 upon endogenous TRPM3 channels, we used pancreatic β-cells of the line INS-1 cells and examined the effects of the CK2 antagonist, CX-4945, upon the TRPM3-mediated Ca^2+^ entry evoked by PS and CIM0216 (Figure 5A). Similar to HEKα2 cells, INS-1 cells did not show any change of [Ca^2+^]_i_ after direct addition of 10 µM CX-4945 (data not shown) indicating that CX-4945 does not act as a direct agonist of Ca^2+^ entry pathways in these cells.

At a concentration of 10 µM, the TRPM3 agonist, CIM0216, has been shown to activate TRPA1 channels, too [9]. These channels have been proposed to induce basal insulin release from β-cells and may be expressed in INS-1 cells as well [29]. Therefore, we tested next whether co-stimulation of INS-1 cells with 100 µM PS and 1 µM CIM0216 provoked Ca^2+^ entry through any other channel than TRPM3. Three independent TRPM3-deficient INS-1 cell lines, INS-1 M3KO-1, INS-1 M3KO-2, and INS-1 M3KO-3 [4], did not show a significant increase of [Ca^2+^]_cyt_ after the addition of 100 µM PS and 1 µM CIM0216. (Figure 5A). In contrast to these TRPM3 knockout lines, we observed a strong and immediate increase of [Ca^2+^]_cyt_ in wild-type INS-1 cells, demonstrating that, under these condition, Ca^2+^ entry is exclusively induced by TRPM3 channels.

This increase was significantly enhanced after pre-incubation with CX-4945, strongly indicating for one thing that TRPM3 channels in INS-1 β-cells are targets of CK2-mediated phosphorylation and for another that CK2 phosphorylation reduces TRPM3-dependent Ca^2+^ entry. To further support this conclusion, we transfected INS-1 cells with cDNA encoding EGFP–CK2 fusion proteins and compared the TRPM3-mediated Ca^2+^ entry in green fluorescent cells to that in non-fluorescent neighboring cells (Figure 5B). Before stimulation, we observed increased Ca^2+^ levels in cells expressing green fluorescent CK2 fusion proteins. This might be related to increased CK2 phosphorylation of proteins different from TRPM3 in INS-1 cells. In contrast, after the stimulation of TRPM3 channels with PS and CIM0216, we found strongly reduced Ca^2+^ levels in CK2-expressing cells, again suggesting that CK2 phosphorylation inhibits TRPM3 channels in INS-1 β-cells.

## 3. Discussion

### 3.1. TRPM3 Channels Are Controlled by Phosphorylation

Phosphorylation is a common regulatory mechanism to control most cellular pathways, and almost all TRPC and TRPV channels have been described as targets of a number of kinases [30]. However, only little is known about the phosphorylation of TRPM channels. For TRPM7, it has been shown that gating is inhibited by protein kinase A due to the phosphorylation of the residue S_1269_ c-terminal to the coiled coil domain [31]. For TRPM4 channels, the Ca^2+^ sensitivity has been shown to be regulated by protein kinase C–dependent phosphorylation at S_1152_ and S_1145_ within the C-terminus of the protein [32] and that casein kinase 1 (CK1)–mediated phosphorylation of S_839_ is responsible for the basolateral localization of this channel in polarized epithelial cells [33]. Very recently, constitutive phosphorylation of four C-terminal serine residues has been identified as a key regulator of TRPM8 channels [34]. TRPM3 channels, however, have never been described as targets of phosphorylation. 

Here, we show for the first time that TRPM3 channels are controlled by phosphorylation. We found TRPM3 to be phosphorylated by CK2, but TRPM3 might also be the target of other kinases that need to be identified. Among 32 putative CK2 phosphorylation sites present in TRPM3 proteins, we only found one single residue (S_1280_) to be selectively phosphorylated. A comprehensive analysis of the phosphoproteome of murine pancreatic islets and Min6 insulinoma cells [35] as well as INS-1 cells [36] by mass spectrometry of peptides obtained by tryptic digestion did not reveal phosphorylation of this residue. This might be related to the low pI and molecular mass of the S_1280_-containing tryptic peptide HES_1280_DQDER and, therefore, might have been overlooked. Due to our experimental approach of testing short TRPM3-derived peptides in vitro, the identification of S_1280_ does not rule out CK2-phosphorylation of additional TRPM3 residues in vivo. We also observed phosphorylation of a peptide, EKDDRFNS_1322_S_1323_NDERIR, containing two adjacent serine residues, which was undetectable when both residues were exchanged with alanine. However, this phosphorylation appeared to be less prominent and dependent on the presence of both residues since phosphorylation disappeared as soon as one of the two residues was mutated. This is rather unexpected for target sites of CK2 [37,38] and might, therefore, rely on unspecific interaction of CK2 with this peptide.

How the phosphorylation of S_1280_ reduces TRPM3-mediated Ca^2+^ entry remains an open question. An indirect mechanism leading to diminished levels of TRPM3 at the plasma membrane could be a plausible explanation. Similar to TRPM3, CK2 phosphorylates a single serine residue (S_812_) within the C-terminus of TRPP2 channels. This phosphorylation enhances the binding of phosphofurin acidic cluster sorting protein (PACS)-1 and PACS-2 to acidic clusters surrounding the phosphorylation site [39]. Binding of PACS proteins reduces the presence of TRPP2 channels at the plasma membrane due to their retrieval from the Golgi apparatus to the endoplasmic reticulum. Since acidic clusters were identified within many other ion channels and since (among others) TRPV4 channels bind PACS proteins as well, the binding of PACS molecules to phosphorylatable acidic clusters of ion channels has been proposed as an important mechanism for the sorting of ion channels to cellular compartments and the control of their activity [39]. Similar to S_812_ of TRPP2, S_1280_ of TRPM3 is flanked by acidic aspartic and glutamic acid residues and we observed enhanced fluorescence inside the cell after overexpression of TRPM3–EGFP fusion proteins, e.g., in INS-1 cells (unpublished observation). Thus, after phosphorylation of S_1280_, the binding of PACS proteins to TRPM3 channels may account for a reduced presence of TRPM3 channels at the plasma membrane and a decreased TRPM3-mediated Ca^2+^ entry.

On the other hand, a direct interference with the functionality of TRPM3 channels may also explain the effects of S_1280_-phosphorylation upon TRPM3-dependent Ca^2+^ entry. The residue S_1280_ is located in the cytosolic C-terminus of the TRPM3 protein in a domain that has not yet been attributed any specific function (Figure 3A,C). It lies close to the TRP domain that is highly conserved throughout all TRPV, TRPC and TRPM channels [40], pivotal for the gating of TRPV1 channels [41] and a target for the regulation of TRPM8 channels by PIP_2_ [42]. Cryo-electron microscopy of the structurally related TRPM7 channel identified tryptophan (W), lysine (K), and arginine (R) residues of the TRP domain—that are conserved in TRPM3—as interaction sites of *N*-terminal amino acids located immediately before the transmembrane helix S1and within the S4–S5 linker [27]. Based on this finding, the TRP domain and the S4–S5 linker have been proposed to “couple the movement of the voltage sensor like S1 to S4 domains to the channel gate” [27]. Also in close proximity but c-terminal to S_1280_ of TRPM3 lies the connecting helix domain that is equally well conserved in TRPM1/3/6/7 proteins [27] (Figure 3A,C). This domain determines the interaction of the *N*- and C-termini of TRPM7 proteins. Thus, phosphorylation of TRPM3 within a region between the TRP domain and the connecting helix domain might interfere with channel gating and or *N*/C terminal interaction.

S_1280_ is not the subject of alternative splicing and is present in all TRPM3 variants described (Figure 3A, [26]). Therefore, its phosphorylation appears as a common mechanism to control all TRPM3 isoforms. Since we used both PS and CIM0216 for activation, short pore isoforms such as TRPM3α2 as well as long pore isoforms such as TRPM3α1 might be controlled by CK2-mediated phosphorylation in INS-1 cells [10]. In HEKα2 cells however, TRPM3 channels that are exclusively composed of short pore TRPM3α2 channels displayed CK2-dependent control of Ca^2+^ entry. S_1280_ is also well conserved in human and murine TRPM3 proteins (not shown), suggesting similar functions in TRPM3 proteins of different species. Within the TRPM channel family, however, S_1280_ is located within a heterogeneous region. Serine residues at homologous positions are absent in the closest relatives TRPM1 and TRPM6 or are not targets of CK2 in TRPM7 (Figure 3C). Thus, phosphorylation of the residue S_1280_ by CK2 appears to be a specific mechanism for the control of TRPM3 channels. 

### 3.2. Contribution of CK2-Mediated Phosphorylation of TRPM3 to the Control of Insulin Release

A precise regulation of the insulin release from pancreatic β-cells is crucial for glucose homeostasis. Our previous data showed that TRPM3-deficient mice display impaired blood glucose clearance after glucose administration [4]. Accordingly, TRPM3 antagonists attenuated glucose-stimulated insulin secretion from INS-1 β-cells [43] and TRPM3-deficient INS-1 cells showed reduced Ca^2+^ entry and insulin release in response to PS and glucose [4]. Therefore, we proposed a significant role for TRPM3 in the control of glucose-stimulated insulin release (GSIS) [4]. Since inhibition of CK2 increases both cytosolic Ca^2+^ and GSIS [22] and—as we show now—phosphorylation by CK2 inhibits TRPM3-mediated Ca^2+^ entry, it is reasonable to propose that the CK2-mediated inhibition of TRPM3 plays a role in the control of GSIS. This hypothesis is in line with a number of other observations describing CK2 as a general inhibitor of insulin including CK2 phosphorylation of transcription factors to reduce insulin production and the phosphorylation of kinesin heavy chain and muscarinic receptors 3 to inhibit insulin release (reviewed in [44]). Very similar to TRPM3, we identified two serine residues of Cav2.1 channels as targets of CK2 phosphorylation and showed that CK2 inhibition of Cav channels is followed by increased intracellular Ca^2+^ and insulin release [22]. Thus, CK2 phosphorylation may limit Ca^2+^ entry through both channels and may provide a common mechanism to restrict Ca^2+^ entry and insulin release in ß-cells. Accordingly, INS-1 cells displayed enhanced TRPM3-initiated (PS-stimulated) Ca^2+^ entry when CK2 was inhibited by CX-4945 and reduced TRPM3-dependent Ca^2+^ signals when CK2 was overexpressed (Figure 5). However, TRPM3 channels depolarize the membrane and consequently activate Cav channels so that both channels together contribute to Ca^2+^ entry and insulin release after TRPM3 activation [4]. Hitherto, the affinity of CK2 for both channels as well as their relative abundance inside the β-cell is unknown. Hence, the extent of the CK2-dependent inhibition of the individual channels remains an open question, and the data shown in Figure 5 do not allow conclusive statements about the relative contribution of CK2 phosphorylation of Cav and TRPM3 channels to TRPM3-initiated Ca^2+^ influx.

## 4. Materials and Methods

### 4.1. Cell Culture and Reagents

If not otherwise indicated, commercial material was purchased from Fisher Scientific (Schwerte, Germany). HEK293 cells and HEKα2 cells [24] were cultured in Dulbecco’s minimal essential medium at 37 °C in a humidified atmosphere containing 5% CO_2_ and were passaged twice a week. INS-1 cells [45] and the TRPM3-deficient INS-1 lines M3KO-1, M3KO-2, and M3KO-3 [4] were cultured as described [4]. A 10 mM stock solution of the CK2 inhibitor CX-4945 or 100 mM stock solutions of pregnenolone sulfate (PS) and CIM0216 were dissolved in dimethyl sulfoxide (DMSO, Merck, Darmstadt, Germany) and stored at −20 °C. For use, the reagents were diluted in Krebs Ringer bicarbonate HEPES buffer (KRBH; 135 mM NaCl, 3.6 mM KCl, 5 mM NaHCO_3_, 0.5 mM NaH_2_PO_4_, 0.5 mM MgCl_2_, 1.5 mM CaCl_2_, 10 mM HEPES, 0.1% BSA w/v, 290 ± 5 mOsm/L, pH 7.4) to a final concentration as indicated in the figures and compared to controls with DMSO alone.

### 4.2. Mutagenesis of TRPM3, Plasmids, Transfection, and Fluorescence-Activated Cell Sorting (FACS)

The following oligonucleotide primers 5’*GCG* GAC CAG GAC GAA AGG GAC TAC G and 5´CTC ATG CTT CCT CCA CCG GC and the QuickChange site-directed mutagenesis kit (Agilent Technologies, Waldbronn, Germany) were used to exchange serine_1172_ in TRPM3α2 for alanine. The plasmids TRPM3α2-IRESGFP [5] and TRPM3α2 S_1172_A-IRESGFP allowing the bicistronic expression of TRPM3 and the EGFP cDNAs as well as the plasmid pEGFP-CK2α encoding EGFP–CK2α fusion proteins were transfected using Lipofectamine-2000 according to the manufacturer’s instructions. For the experiment shown in Figure 4A, TRPM3-expressing cells were identified by the green fluorescence of the co-expressed green fluorescent protein (GFP) and isolated by fluorescence-activated cell sorting on a MoFlo XDP cell sorter (Beckman Coulter).

### 4.3. Fura-2 Ca^2+^ Imaging

Ca^2+^ imaging experiments were performed essentially as described [22]. In brief, 2–3 days after seeding on poly-L-lysine-coated glass coverslips, cells were loaded for 30 min at 37 °C and 5% CO_2_ with 5 µM fura-2-acetoxymethyl ester (Fura-2AM)dissolved in KRBH buffer. In some experiments (Figure 2A and Figure 5A), the solution also contained 10 µM CX-4945 or solvent only (DMSO, control). Cells were washed twice and analyzed for 1 min in 300 µL KRBH buffer at room temperature by fluorescence excitation every 3 s at wavelengths (λ) of 340 and 380 nm and detection of fluorescence emissions at λ > 510 nm. PS and CIM0216 (or solvent only as control) dissolved in 300 µL KRBH buffer as were added as indicated in the figures.

### 4.4. In Vitro Phosphorylation of Immunopurified Proteins

HEK293 cells and HEK293 cells transiently transfected with the TRPM3α2 cDNA were grown in 75 cm^2^ flasks to ~80% confluency, washed with phosphate buffered saline, and frozen at −80 °C. Proteins were extracted at 4 °C with 1 mL solubilization buffer containing 100 mM HEPES, pH 7.5, 150 mM NaCl, 1 mM CaCl_2_, 1% digitonin, and supplemented with cOmplete™ ULTRA Tablets EDTA-free protease inhibitors (Roche, Mannheim, Germany). Cell debris was removed by centrifugation (100,000× *g*, 4 °C), and extracts were subjected to immunoprecipitation. For that purpose, 100 µL protein G Sepharose beads were equilibrated thrice in buffer before cell extracts were precleared over a period of 1 h at 4 °C. The supernatant was incubated with 10 µg of affinity-purified rat monoclonal anti-TRPM3 antibodies [8] or polyclonal TRPM3 antibodies [4] for 1 h at 4 °C and subsequently added to another aliquot of equilibrated protein G Sepharose. Immunoprecipitation was continued overnight with gentle shaking at 4 °C, before the beads were washed three times. Bound proteins were resuspended in 20 µL kinase buffer (50 mM Tris-HCl, pH 7.5, 100 mM NaCl, 10 mM MgCl_2_, 1 mM dithiothreitol, 50 µM ATP) and incubated at 37 °C for 30 min with 185 kBq [γ-^32^P]ATP or [γ-^32^P]GTP (Hartmann Analytic, Braunschweig, Germany) in the presence or absence of 0.4 µg purified, recombinant CK2 (0.4 µg/µL) [46]. Nucleolin served as a control substrate in a separate reaction. The reactions were stopped by the addition of 3× SDS sample buffer (195 mM Tris-HCl, pH 6.8, 6% SDS, 15% β-mercaptoethanol, 30% glycerol, 0.03% bromophenol blue) at 95 °C for 5 min. Finally, proteins were separated in a 7.5% SDS polyacrylamide gel. The gel was dried under vacuum and analyzed by autoradiography (Carestream^®^ X-Omat LS film, Sigma-Aldrich, Munich, Germany).

For immunoblot analysis of phosphorylated protein with a phospho-CK2 substrate motif antibody (Cell Signaling #8738), extraction, immunoprecipitation, phosphorylation using GTP, and SDS PAGE were performed as described above. Proteins were blotted to a PVDF membrane, and the membrane was incubated in Tris-buffered saline (TBS, 20 mM Tris-HCl, pH 7.4, 150 mM NaCl) with 0.1% Tween20 (TBS-T) and 5% bovine serum albumin (BSA) for one hour at room temperature. Incubation with the phospho-CK2 substrate motif antibody was performed in a dilution of 1:1000 in TBS-T/5% BSA at 4 °C overnight. The membrane was washed twice for 10 min with TBS-T, incubated with a peroxidase-coupled secondary anti-rabbit antibody for 1 h at room temperature and washed again twice for 10 min with TBS-T before phosphorylated proteins were visualized using the SuperSignal™ West Femto Maximum Sensitivity Substrate. 

### 4.5. In Vitro Phosphorylation of a TRPM3 Peptide Library

Pentadecapeptides were synthesized on polyethlylene-glycol-derivatized cellulose membranes (amino-PEG500-UC540, Intavis, Cologne, Germany) using an Intavis ResPepSL peptide spot synthesizer. The filters were equilibrated in kinase buffer (50 mM Tris-HCl, pH 7.5, 100 mM NaCl, 10 mM MgCl_2_, 1 mM dithiothreitol, 50 µM ATP) supplemented with 1% bovine serum albumin (BSA) overnight at 4 °C, incubated for one hour at room temperature in 2 mL kinase buffer containing 740 kBq [γ-^32^P]ATP in the presence or absence of recombinant CK2 before they were washed three times with kinase buffer containing 1 M NaCl. Bound proteins were removed with 8 M urea, 35 mM SDS, and 1% 2-mercaptoethanol. Finally, the membranes were washed with ethanol and dried. Phosphorylated peptides were visualized using autoradiography.

## Figures and Tables

**Figure 1 ijms-22-13133-f001:**
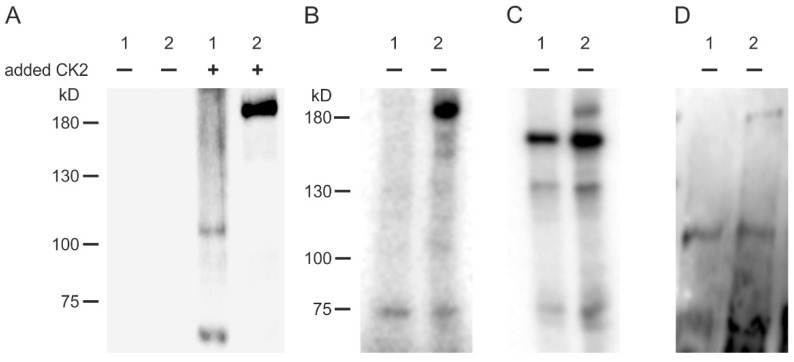
CK2 phosphorylates TRPM3 channels. Immunoprecipitated proteins from HEK293 cells (1) and TRPM3α2-expressing HEK293 cells (2) were incubated in the presence (+) or absence (-) of added CK2 and separated by gel electrophoresis. (**A**) Autoradiograph (short exposure, ~2 h) of proteins that were immunoprecipitated using monoclonal anti-TRPM3 antibodies and phosphorylated in the presence of [γ-^32^P]ATP. (**B**) The same as in (**A**) but in the presence of [γ-^32^P]GTP and the absence of added CK2 (long exposure, ~24 h). (**C**) Same as in (**B**), but proteins were immunoprecipitated using polyclonal anti-TRPM3 antibodies. (**D**) Same as in (**C**), but separated proteins were blotted onto a PVDF membrane and analyzed with a phospho-CK2 substrate motif antibody.

**Figure 2 ijms-22-13133-f002:**
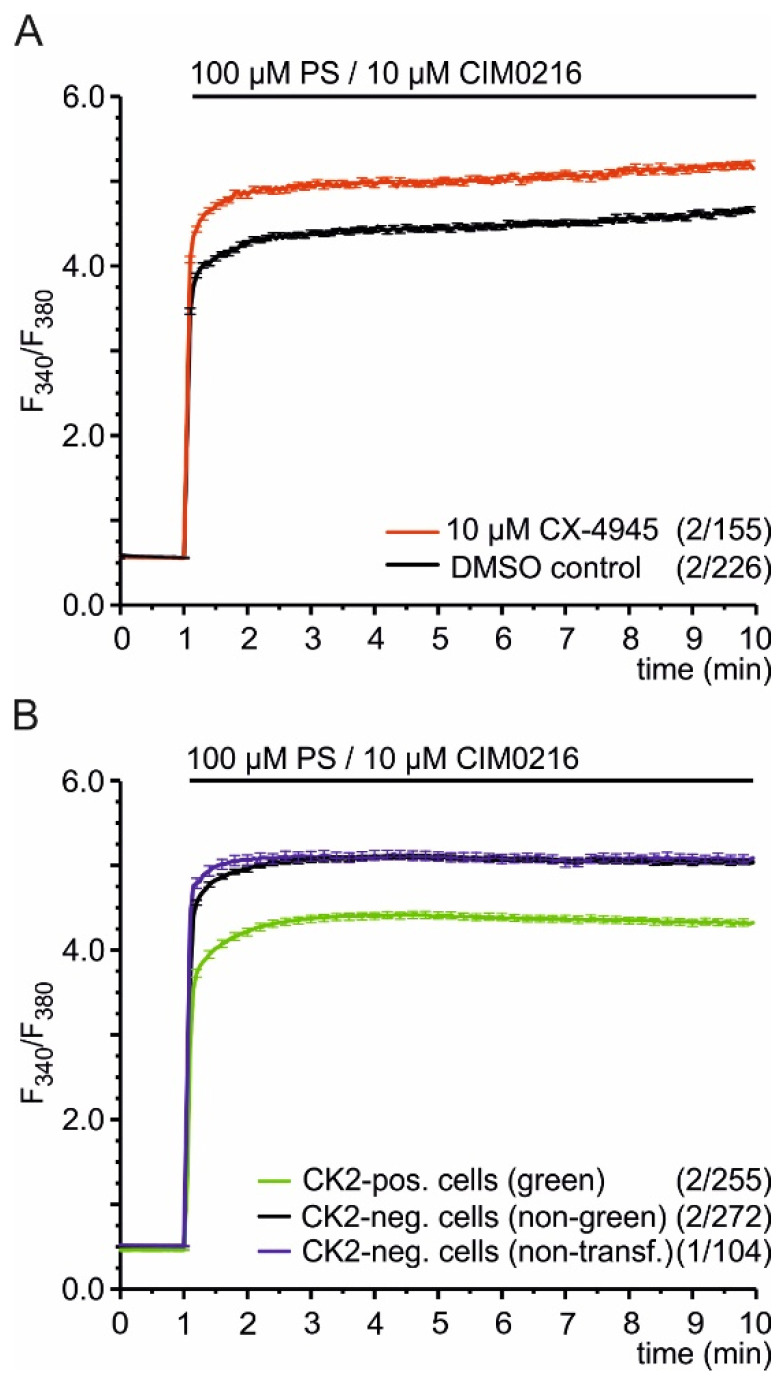
TRPM3-mediated Ca^2+^ signals are inhibited by CK2. (**A**) Ca^2+^ signals induced by PS/CIM0216 in stable TRPM3α2-overexpressing HEK293 cells (HEKα2) in the presence (red trace) or absence (black trace, solvent DMSO only) of the CK2 inhibitor CX-4945. Cells were pre-incubated for 30 min in CX-4945 or DMSO. (**B**) Fura-2 Ca^2+^ imaging experiments with HEKα2 cells transfected with cDNA encoding EGFP–CK2α fusion proteins and stimulated with pregnenolone sulfate (PS)/CIM0216. Ca^2+^ signals of green fluorescent cells (green) were compared to non-green cells (black) of the same dish as well as to non-transfected cells (blue). The number of independent experiments and the total number of analyzed cells are each indicated in brackets.

**Figure 3 ijms-22-13133-f003:**
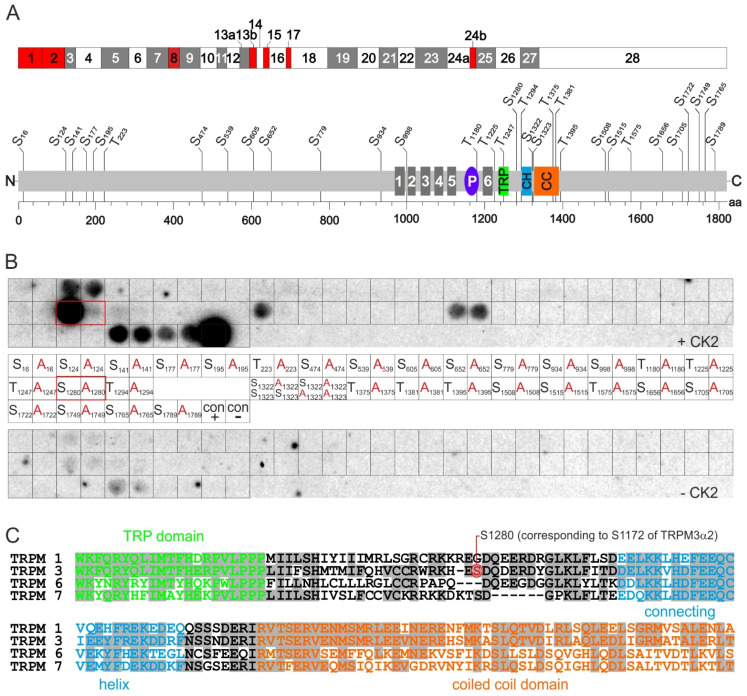
CK2 phosphorylation of a single serine residue of TRPM3. (**A**) Localization of putative CK2 phosphorylation sites within TRPM3 proteins. The mouse *Trpm3* gene comprises 28 exons (upper panel, [26]). Alternative *N*-termini of α-isoforms and β-isoforms are encoded by exon 1 and exon 2, respectively (shown in red). Likewise, as a result of alternative splicing, the protein regions encoded by exons 8, 13b, 15, 17, and 24b (shown in red) are absent in several isoforms. The organization of domains of the encoded TRPM3 proteins (light gray bar) is shown below true to scale. Well-conserved domains such as transmembrane helices S1–S6 (dark gray), channel pore (P, violet) TRP domain (TRP, green) a *N*/C-terminal connecting helix (CH, blue), and a coiled coil domain (CC, orange) are indicated [15]. The positions of 23 serine (S) and 9 threonine (T) amino acid residues (aa), representing putative CK2 phosphorylation sites, are indicated by numbers. (**B**) Analysis of all putative CK2 phosphorylation sites within TRPM3 reveals selective phosphorylation of S_1280_. In all, 15 mer-peptides each comprising one of the 32 putative phosphorylation sites (S/T XX D/E) within the complete TRPM3 amino acid sequence (including all known exons) were synthesized on cellulose membranes. Corresponding peptides carrying S/T to A mutations were synthesized next to them, each. Membranes were incubated with [γ-^32^P]ATP in the presence (+CK2) or absence (-CK2) of protein kinase CK2. Peptides of the sequence RRRDDDSDDD/RRRDDDADDD served as positive (con+) and negative (con-) controls, respectively. Note the selective phosphorylation of S_1280_ (highlighted by a red rectangle). (**C**) S_1280_ (highlighted in red, corresponding to S_1172_ of TRPM3α2) is not conserved within TRPM proteins as shown by sequence alignment of mouse TRPM3 (NP_001030319.1) with its closest relatives TRPM1 (NP_001034193.2), TRPM6 (NP_700466.1), and TRPM7 (NP_067425.2). Amino acid residues identical to TRPM3 are highlighted by gray backgrounds. Amino acid residues belonging to the TRP-domain [27], the connecting helix [27], and the coiled coil domain [28] are labeled in green, blue, and orange, respectively.

**Figure 4 ijms-22-13133-f004:**
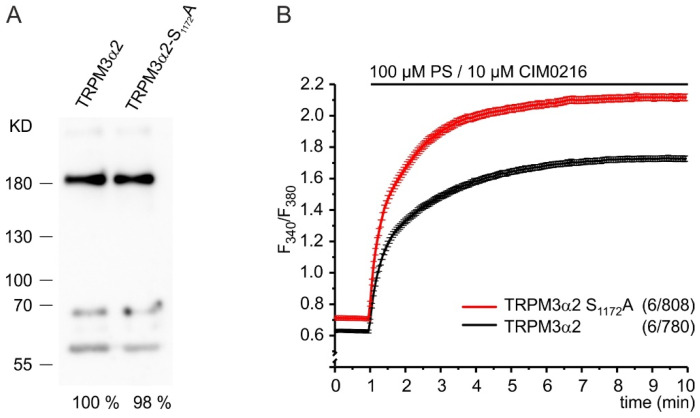
Replacement of serine residue 1172 with alanine strongly enhances TRPM3-mediated Ca^2+^ entry. (**A**) Western blot analysis using monoclonal anti-TRPM3 antibodies of immunoprecipitated proteins from equal numbers of FACS-sorted cells transfected as described in (B). Densitometric analysis of signals at the appropriate molecular mass for TRPM3 revealed relative signal intensities (%) as indicated beneath the blot. (**B**) Fura-2 Ca^2+^ imaging experiments of green fluorescent HEK293 cells analyzed 48 h post transfection of plasmids allowing the bicistronic expression of the green fluorescent protein and TRPM3α2 (black trace) or TRPM3α2 S_1172_A (red trace), respectively. The duration of the presence of pregnenolone sulfate (PS) and CIM0216 is shown. The number of independent experiments and the total number of analyzed cells are indicated in brackets.

**Figure 5 ijms-22-13133-f005:**
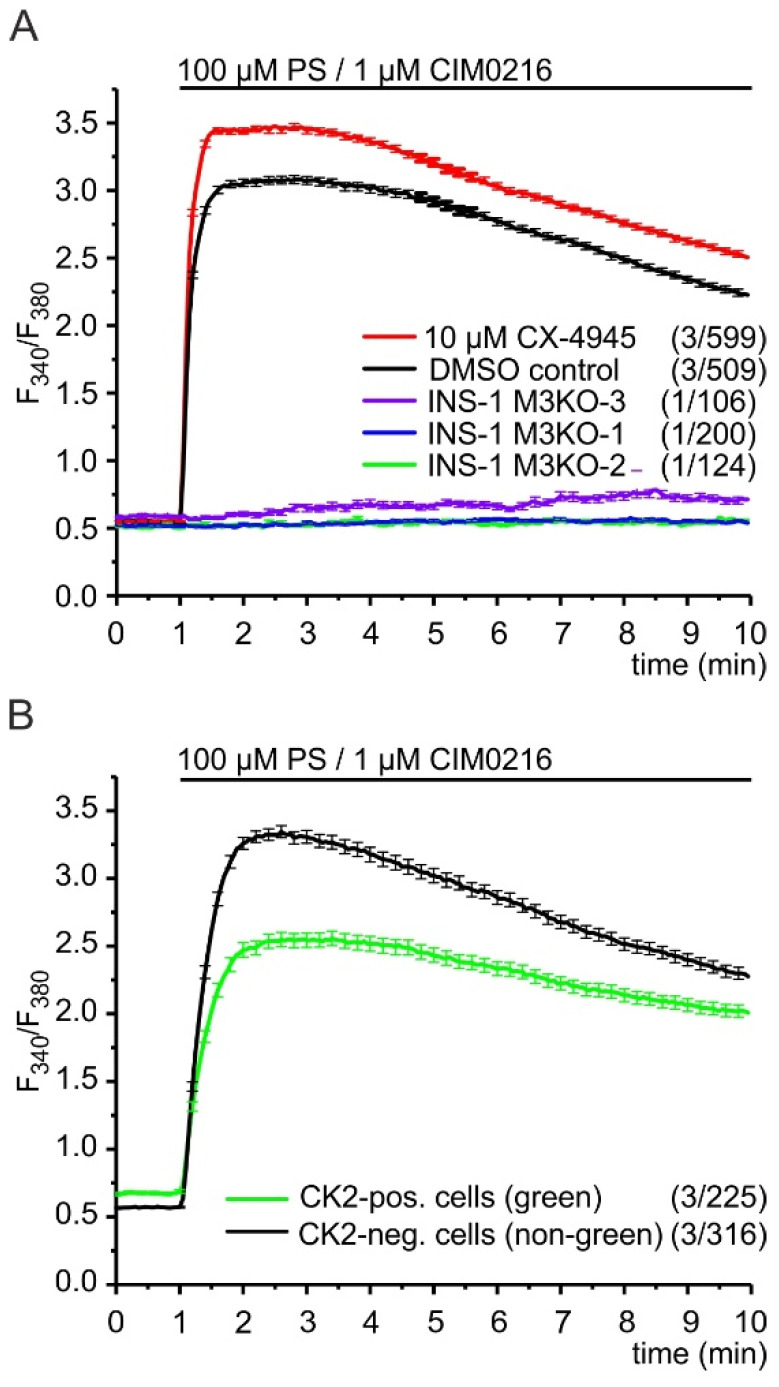
Protein kinase CK2 controls TRPM3-mediated Ca^2+^ signals in INS-1 β-cells. (**A**) The pregnenolone sulfate (PS)/CIM0216-induced Ca^2+^ entry in INS-1 cells was exclusively induced by TRPM3 and increased after treatment with the CK2 inhibitor, CX-4945. Cells from three independent TRPM3-deficient INS-1 cell lines (green, blue, and violet traces) did not respond to the TRPM3 agonists PS (100 µM) and CIM0216 (1 µM). In contrast, wild-type INS-1 cells that were pre-incubated for 30 min in 10 µM CX-4945 (red trace) showed increased PS/CIM0216-induced Ca^2+^ signals compared to that in control cells treated with solvent (DMSO) only (black trace). (**B**) Pregnenolone sulfate (PS)/CIM0216-induced Ca^2+^ signals in INS-1 cells were reduced after the introduction of recombinant CK2α. INS-1 cells were transfected with pEGFP-CK2α. Green fluorescent cells expressing EGFP–CK2α fusion proteins (green trace) were compared to non-green cells of the same dish (black trace). The number of independent experiments and the total number of analyzed cells are indicated in brackets, each.

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
