# Peer review of "Control of TRPM3 Ion Channels by Protein Kinase CK2-Mediated Phosphorylation in Pancreatic β-Cells of the Line INS-1"

_ijms, 2021, doi:10.3390/ijms222313133_

Round 1

Reviewer 1 Report

Dear Editor and Authors,

this study by Becker et al. convincingly demonstrates the role of TRPM3 phosphorylation by CK2 in the function of beta cells of the Langerhans islets. The authors used a specific CK2 inhibitor and CK2 overexpression to show that CK2 activity inhibits pregnenolone-sulphate (PS)- and CIM0216 -induced Ca2+ influx in HEK293- and INS-1 cell lines, expressing TRPM3. There are however several concerns that need to be addressed.

Major concerns

  • Figure 4. shows that the Ca2+ signal was higher in HEK293 cells expressing the S1172A TRPM3. These data, however, do not demonstrate that this effect would be CK2 dependent, as the phosphorylation site might be a target of other kinases too. To confirm the role of S1172 in CK2 phosphorylation, further experiments are required to show that CX-4945 does not influence the Ca2+ signal amplitude in S1172A TRPM3 expressing HEK293 cells. For the same purpose, additional experiments could be performed on S1172A TRPM3 HEK293 cells, overexpressing CK2.
  • Results shown in Figure 4 could also be explained by a higher expression of the mutant channel, or by different gating properties or cation permeability. These possibilities should be checked.
  • Ca2+ imaging in Figure 5. was performed using INS-1 cells, which co-express TRPM3 and L-type Ca2+ channels (LTCC). Therefore, the PS and CIM0216-induced Ca2+ signals measured here was a result of Ca2+ entry through both channel types (as TRPM3-mediated depolarization induces LTCC opening). Since LTCCs were found to be CK2 targets, the enhanced Ca2+ signal of CX-4945 cells possibly be due to the increased activity of LTCCs (as a direct result of dephosphorylation of LTCCs or alternatively, because of higher depolarization, caused by hyperactive TRPM3 channels). In order to rule out these LTCC-related effects and confirm the significant role of TRPM3 phosphorylation in the process, similar experiments should be performed in the presence of an LTCC blocker.

Minor comments

     The following questions should be discussed in the manuscript:

  • What intracellular factors are involved in TRPM3 activation during glucose stimulation?
  • What determines CK2 activity in beta cells? Where does basal, tonic activity arise from?
  • Is CK2 inhibition a potential therapeutic strategy in diabetes?
  • In Materials and Methods: how the TRPM3 KO cell lines were generated?

Reviewer 2 Report

General remarks:

Becker et al. report here a straight-forward continuation of their previous studies to analyze the mechanism of CK2-mediated negative feedback regulation of beta cells. The authors now extend their previous concept of CK2-mediated, inhibitory phosphorylation of beta cell Ca2+ entry channels, which was established for CaV2.1, by a similar regulatory mechanism for TRPM3. The authors show convincingly the ability of CK2 to phosphorylate TRPM3 und identify a potentially involved regulatory phosphorylation site. In comparison to their previous studies, the present report appears less convincing for reasons outlined below. Specifically, some obvious experimental tests of the proposed concept as well as controls are missing.

Major points:

  • Why did the authors decide to use a combination of PS plus CIM0216 in their tests for a CK2-dependent, negative feedback regulation of TRPM3? The rationale of this experimental design hardly clear. This activation protocol must be associated with excessive, cellular Ca2+ loading, as also evident from the Ca2+ imaging experiments. As shown previously for CaV2.1, CK2 is expected to physically interact witin a complex with its channel substrate to perform the negative feedback regulation. If this is the case also for TRPM3, then local Ca2+ entry should be sufficient for the process. Is the regulation also evident at lower level of TRPM3 activity (activated with PS only)?
  • Along the same lines, do the authors have any evidence for association between TRPM3 and CK2?
  • As to the Ca2+ signals recorded in TRPM3 overexpressing HEK cells, these signals are a bit difficult to interpret since the overall TRPM3 expression level is obviously much higher in the stable as compared to the transient expression situation. This is evident from the marked differences in Ca2+ signal kinetics (Figure 2 vs Figure 4). Therefore the stoichiometry of a channel – kinase interaction may be quite different. This aspect might deserve attention in the discussion. Moreover, in both setting it would be indeed important to include the responses of the cell lacking overexpression (control lines/ sham transfected HEK cells).
  • It is unclear why the authors conclude that “ …the (PS)/CIM0216-induced Ca2+ entry in INS-1 cells is exclusively mediated by TRPM3”(Figure 5). IN view of the rather plausible, tight interaction/cooperation between CaV2.1 and TRPM3, PS-induced Ca2+ signals may well involve entry through CaV,and this contribution is not at all inconsistent with the lack of (PS-induced) function phenotype of the TRPM3-KO cells. Please clarify in the discussion.

It is suggested to corroborate the relevance of the identified serine, which is suggested as the basis of the feedback regulation in a set of experiments with a phospho-mimicking mutation.

Reviewer 3 Report

In this manuscript, Becker et al. present a new control mechanism in which the TRPM3 ion channel activity is partially inhibited due to phosphorylation by the CK2 kinase. They identify the phosphorylated residue and show functional changes in ion channel activity in several cell lines. This form of activity modulation by TRPM3 was previously unknown and is interesting on its own. The work is a direct extension of previous work by Becker and Philipp where (2020 Cell Physiol and biochem) they show a function of TRPM3 ion channels and their knockout in a celline resembling pancreatic beta cells and (2020 Int. J. Mol. Sci) they show the involvement of the CK2 kinase in mediating calcium signals in beta cells due to CaV2.1 channel activity modulation. 

The data presented in the current manuscript seems gathered according to the current high standards in the field, and the experiments are largely appropriate and support the drawn conclusions. It is a bit disappointing to not find any direct electrophysiology patch clamp measurements in a paper describing a novel activity modulating system of an ion channel. 

I do, however, have some remarks that might benefit this manuscript before publication. 

1) The title indicates "pancreatic beta cells" however, nowhere in the manuscript were beta cells used. Please indicate correctly in the title and throughout the manuscript you used "beta-cell derived INS-1 cells". Or even better; provide a proof of concept for your experiment in primarily isolated pancreatic beta cells.

2) Is there a way you can exclude a direct effect of CX-4945 on TRPM3? Maybe the possibility to reduce CK2 production or activity on the genetic level? In the current experiments, there seems to be a pre-incubation of CX4945 with the cells, what happens when you add cx4945 directly on TRPM3 either in combination with PS or CIM or not? Are the effects of CX4945 exactly the same as quinalizarin, in this case, an off-target effect directly on TRPM3 would be less likely to interfere with the results. 

3) The way of creating the 15-mer peptides and systematically checking their phosphorylation is an excellent approach. The S1280 is clearly identified as a phorphorylated residue. The S1322/1323 however seems to be more ambiguous. Mutating any of the residues can render the neigboring residue inaccessible and such. Maybe pay a bit more attention in the discussion/results that the S1280 might not be the only residue responsible to exert the CK2-mediated activity modulation of TRPM3. 

4) In figure 5a, there seems to be huge difference between the phosphorylated TRPM3 and the M3KO cellines. It is obvious that TRPM3 inhibition due to phosphorylation is incomplete. Could you extend these experiments with a TRPM3 inhibitor like isosakuranetin? 

5) Can you expand your FURA2 imaging. The paper you use as 'previously described' clearly indicates different experiments as far as the used buffers and timing of the experiment are concerned. At this moment it is unclear if you used glucose in the solutions for the INS1 cell measurements, and if so, what concentration. 

Best of luck with your future research. 

Round 2

Reviewer 1 Report

The authors answered all my questions and dealt with my concerns, thus i recommend the publication of this manuscript in its current form.

Author Response

We thank the reviewer for his positive evaluation and his support. 

Reviewer 2 Report

The authors have done an excellent job to amend the manuscript. The revised version is valuable contribution to the field.

Author Response

(The authors gave the same response as above.)
